# Assessment of Renal Allograft Stiffness and Viscosity Using 2D SWE PLUS and Vi PLUS Measures—A Pilot Study

**DOI:** 10.3390/jcm11154370

**Published:** 2022-07-27

**Authors:** Felix-Mihai Maralescu, Felix Bende, Ioan Sporea, Alina Popescu, Roxana Șirli, Adalbert Schiller, Ligia Petrica, Tudor Voicu Moga, Ruxandra Mare, Iulia Grosu, Flaviu Bob

**Affiliations:** 1Department of Internal Medicine II-Division of Nephrology, “Victor Babeș” University of Medicine and Pharmacy, EftimieMurgu Sq. No. 2, 300041 Timișoara, Romania; felixmihai21@gmail.com (F.-M.M.); schiller.adalbert@gmail.com (A.S.); ligia_petrica@yahoo.co.uk (L.P.); iuliag_13@yahoo.com (I.G.); flaviu_bob@yahoo.com (F.B.); 2Centre for Molecular Research in Nephrology and Vascular Disease, Faculty of Medicine, “Victor Babeș” University of Medicine and Pharmacy, EftimieMurgu Sq. No. 2, 300041 Timișoara, Romania; 3Department of Gastroenterology and Hepatology, “Victor Babes,” University of Medicine and Pharmacy, 300041 Timișoara, Romania; isporea@umft.ro (I.S.); alinamircea.popescu@gmail.com (A.P.); roxanasirli@gmail.com (R.Ș.); moga.tudor@yahoo.com (T.V.M.); ruxandra.mare@gmail.com (R.M.); 4Advanced Regional Research Center in Gastroenterology and Hepatology, “Victor Babes,” University of Medicine and Pharmacy, 30041 Timișoara, Romania

**Keywords:** chronic kidney disease, stiffness, viscosity, 2D SWE PLUS, Vi PLUS

## Abstract

Elastography is a useful noninvasive tool for the assessment of renal transplant recipients. 2D-shear wave elastography (SWE) PLUS and viscosity plane-wave ultrasound (Vi PLUS) have emerged as novel techniques that promise to offer improved renal stiffness and viscosity measures due to improved processing algorithms. **Methods**: We performed a cross-sectional study of 50 kidney transplanted patients (16 women, 34 men; mean age of 47.5 ± 12.5; mean estimated glomerular filtration rate (eGFR) estimated by Chronic Kidney Disease Epidemiology Collaboration formula: 52.19 ± 22.6 mL/min/1.73 m^2^; and a mean duration after transplant of 10.09 ± 5 years). For every patient, we obtained five valid measurements of renal stiffness (obtained from five different frames in the cortex of the renal graft), and also tissue viscosity, with a C6-1X convex transducer using the Ultra-Fast™ software available on the Aixplorer Mach 30 ultrasound system (Supersonic Imagine, Aix-en-Provence, France). The median values of elastographic and viscosity measures have been correlated with the patients’ demographic, biological, and clinical parameters. **Results**: We obtained a cut-off value of renal cortical stiffness of <27.3 kiloPascal(kPa) for detection of eGFR < 60 mL/min/1.73 m^2^ with 80% sensitivity and 85% specificity (AUC = 0.811, *p* < 0.0001), a cut-off value of <26.9 kPa for detection of eGFR < 45 mL/min/1.73 m^2^ with 82.6% sensitivity and 74% specificity (AUC = 0.789, *p* < 0.0001), and a cut-off value of <23 kPa for detection of eGFR < 30 mL/min/1.73 m^2^ with 88.8% sensitivity and 75.6% specificity (AUC = 0.852, *p* < 0.0001). We found a positive correlation coefficient between eGFR and the median measure of renal cortical stiffness (r = 0.5699, *p* < 0.0001), between eGFR the median measure of viscosity (r = 0.3335, *p* = 0.0180), between median depth of measures and renal cortical stiffness (r = −0.2795, *p* = 0.0493), and between median depth of measures and body mass index (BMI) (r = 0.6574, *p* < 0.0001). Our study showed good intra-operator agreement for both 2D SWE PLUS measures—with an intraclass correlation coefficient (ICC) of 0.9548 and a 95% CI of 0.9315 to 0.9719—and Vi PLUS, with an ICC of 0.8323 and a 95% CI of 0.7457 to 0.8959. The multivariate regression model showed that 2D SWE PLUS values were associated with eGFR, Vi PLUS, and depth of measures. **Conclusions**: Assessment of renal allograft stiffness and viscosity may prove to be an effective method for identifying patients with chronic allograft injury and could prove to be a low-cost approach to provide additional diagnostic information of kidney transplanted patients.

## 1. Introduction

The most prevalent cause of renal graft failure following transplantation is chronic allograft injury [1]. The clinical signs of the malfunctioning renal graft are represented by either edema and an increase in blood pressure. A suspicion is proven by the finding of increased serum creatinine or the presence of proteinuria. The “gold standard” in diagnosing chronic allograft injury or acute rejection is, however, renal allograft biopsy. The present debate is not about whether protocol biopsies provide information about the sub-clinically progressing disease, but rather whether the benefit outweighs the risk.

It is critical to detect an early decline in the eGFR and properly track the disease’s course using a variety of markers (biomarkers, histology, imaging). Traditional ultrasound is used to assess the kidneys; however, this technology only provides us with a few pieces of quantitative information about the renal size and parenchyma thickness, which both decrease as chronic kidney disease (CKD) progresses [2]. A non-invasive method to assess the progression of progressive fibrosis after transplantation would be the simplest solution to this problem. This noninvasive technology should ideally be accurate for fibrosis grading, simple to use, dependable, and affordable, allowing for long-term monitoring of fibrosis progression following transplantation [3].

Ultrasound-based elastography approaches have become well-accepted non-invasive evaluation tools in the differential diagnosis of benign and malignant superficial organs such as breast and thyroid, [4] and for liver fibrosis and cirrhosis [5].

Transient elastography (TE) (FibroScan, EchoSens, Paris, France) was the first and most widely used technology for assessing renal allograft nephropathy. [6,7,8,9], followed by virtual touch quantification (VTQ) [10,11] and 2D-SWE [12].

By detecting the velocity of shear waves produced by a focused ultrasonic beam, 2D SWE PLUS enables a real-time qualitative and quantitative tissue elasticity evaluation. The initial 2D SWE method was developed by Supersonic Imagine (SSI), and multiple research and meta-analyses, confirmed its utility in assessing fibrosis [13,14,15,16].

This study aimed to evaluate the feasibility and the performance of this new ultrasound-based technique (ShearWave Elastography and Viscosity Plane-wave UltraSound) embedded in the new Aixplorer Mach 30 system (Aixplorer, Supersonic Imagine, Aix-en-Provence, France), for the non-invasive assessment of renal allograft fibrosis and viscosity.

## 2. Materials and Methods

### 2.1. Study Population

A monocentric cross-sectional study was performed during a 3-month interval (March 2022 to May 2022) in a tertiary department of nephrology. Ultrasound-based measurements were performed in all patients, in the same session, using ShearWave Elastography (2D SWE PLUS) and Viscosity Plane-wave Ultrasound (Vi PLUS) from Aixplorer (Supersonic Imagine, Aix-en-Provence, France).

The study was approved by the Research Ethics Committee and the Institutional Review Board of our University (number 41/04.03.2022) and was performed following the World Medical Association Declaration of Helsinki, revised in 2000, Edinburgh. All the patients provided written informed consent before study entry.

Fifty kidney transplanted patients (16 women and 34 men, with a mean age of 47.5 ± 12.5, a mean eGFR of 52.19 ± 22.6 mL/min/1.73 m^2^, and a mean duration after transplant of 10.09 ± 5 years) were enrolled after obtaining informed consent.

From every patient we obtained the following data from medical records: age, gender, height, weight, BMI, transplanted kidney length, time from transplant, type of transplant (received from a deceased or living donor), immunosuppressive treatment, and history of hypertension, diabetes, or previous glomerular disorders.

For every patient, blood specimens were obtained, and the following assessments were performed: complete blood count, urea, serum creatinine, sodium, potassium, calcium, phosphorus, parathyroid hormone, uric acid, ESR, C reactive protein, cholesterol, triglycerides, aspartate aminotransferase (AST), alanine aminotransferase (ALT), and total bilirubin. The patients were instructed to collect urine specimens for proteinuria/24 h and urine culture.

### 2.2. ShearWave PLUS Elastography

2D SWE PLUS measurements were performed with a C6-1X convex transducer using the UltraFast™ software available on the Aixplorer Mach 30 ultrasound system (Supersonic Imagine, Aix-en-Provence, France). The Young’s modulus (YM) of the region of interest (ROI) was calculated with the apparatus’s software using the formula E = ρ × cs2, where E is tissue elasticity (in kPa), ρ is tissue density (in kg/m^3^), and cs is shear wave velocity (in m/s) [17]. A quantitative map of tissue stiffness is presented using ultrafast imaging techniques, with a color scale ranging from dark blue to yellow, then dark red, corresponding to YM values ranging from 0 to >50 kPa [18].

Measurements were performed for the middle portion of the renal graft, in the subcapsular cortex, with the patient in dorsal decubitus immediately after emptying the bladder. For each measurement (with ROI set by the renal software system at 10 mm displayed as a Q-box on the screen), the software of the apparatus provided the following information: the mean value of YM—the weighted arithmetic mean of all values in the ROI (displayed as Mean); the minimum value of the YM in the ROI (displayed as Min on the screen); the maximum value of YM in the ROI (displayed as Max on the screen); the standard deviation of the YM values in the ROI (displayed as SD); the sum of all Vi PLUS values in the ROI divided by the number of values (displayed as Vi PLUS Mean); the median values of Vi PLUS (displayed as Vi PLUS Med); the standard deviation of the Vi PLUS values in the ROI (displayed as Vi PLUS SD); the distance between the skin and the examined cortical region (=kidney depth, displayed as Depth in cm); and the diameter of the ROI (displayed as Diam in mm) (Figure 1).

For every patient, we obtained 5 valid measurements of renal stiffness (obtained from 5 different frames in the cortex of the renal graft), and also tissue viscosity blinded to the patients’ medical information. The median value of the five elastographic measurements was correlated with the demographic and clinical parameters of the patients.

After selecting the best acoustic window by ultrasound examination and obtaining a suitable image, the shear-wave measurement box was positioned in the mid-portion of the renal parenchyma, under the renal capsule. Acquisitions were performed during neutral respiratory apnea. After the 2D SWE PLUS map was deemed appropriate, image acquisition was performed, and then, the Q-Box was positioned in an area of relative uniform elasticity, at a depth of 2–4 cm. The median value of five 2D SWE PLUS measures (from 5 different frames), expressed in kPa, was considered an indication of renal cortical stiffness.

Five tissue viscosity measures are expressed in Pascal-seconds (Pa.s). The information on tissue shear wave dispersion (analysis of shear wave propagation velocity at various frequencies) can be displayed using Vi.PLUS. The degree of the shift in shear wave speed across frequencies is qualitatively depicted in a color-coded graphic and numerically expressed in Pa.s throughout a range of values. Vi.PLUS was used in conjunction with the 2D-SWE mode acquisitions and took place at the same time and followed the same technique.

### 2.3. Statistical Analysis

MedCalc Version 19.4 (MedCalc Software Corp., Brunswick, ME, USA) and Microsoft Office Excel 2019 were used for the statistical analysis (Microsoft for Windows). For demographic, anthropometric, and laboratory findings, descriptive statistics were employed. The distribution of numerical variables was determined using the Kolmogorov–Smirnov test. Numerical variables having a normal distribution are represented by means and standard deviation, whereas non-normal distribution variables are represented by median values and range. Percentages and figures were used to represent qualitative variables. The Pearson or Spearman correlation coefficient was used to express correlations between variables. A “*p*” value of <0.05 was considered to be statistically significant. Areas under receiver operating characteristic curves (AUC) were determined for 2D SWE PLUS to identify the optimal cut-off points that maximized the Youden index for staging CKD by eGFR. Positive predictive value (PPV—defined as the ratio of true positive cases to all positive cases), negative predictive value (NPV—defined as the ratio of true negative cases to all negative cases), and diagnostic accuracy (defined as the ratio of true positive and true negative cases to the total number of cases) were calculated. Univariate and multivariate statistical analyses were also implied.

## 3. Results

There were 16 women and 34 men among the 50 transplanted patients included in the study. They had a mean age of 47.5 years, a mean BMI of 27.8, and a mean eGFR of 52 mL/min/1.73 m^2^. Twenty received their new kidney from a living related person and 30 from a deceased donor. The mean time elapsed after transplant was 10.5 years. Thirty-seven out of them had a history of hypertension, 8 of diabetes, and 19 of previous glomerular disorders. The arithmetic mean of the median of the five measures using 2D SWE PLUS was 25.95 kPa, and the arithmetic mean of the median of the five measures using Vi PLUS was 2.82 Pa.s. We found a positive correlation coefficient between eGFR and the median measure of renal cortical stiffness (r = 0.5699, *p* < 0.0001), between eGFR and the median measure of viscosity (r = 0.3335, *p* = 0.0180) (Figure 2 and Figure 3), between the median depth of measures and renal cortical stiffness (r = −0.2795, *p* = 0.0493), and between median depth of measures and BMI (r = 0.6574, *p* < 0.0001) (Figure 4).

Our study showed good intra-operator agreement for 2D SWE PLUS measures with an ICC of 0.9548 and a 95% CI of 0.9315 to 0.9719, and for Vi PLUS, with an ICC of 0.8323 and a 95% CI of 0.7457 to 0.8959 (Figure 5).

As the CKD stage advances, our results indicate a clear decline in mean 2D SWE PLUS measures. Table 1 and Figure 6 both highlight this fact.

No statistically significant correlations were found between mean measures of cortical stiffness and height, weight, BMI, time from transplant, received from a living related or deceased donor, kidney length, hemoglobin, hematocrit, urea, uric acid, cholesterol, triglycerides, ALT, AST, total bilirubin, sodium, potassium, C reactive protein, age, hypertension, diabetes, previous glomerular disease, or if the kidney was attained from a living related or a deceased donor. (Table 2)

We obtained a cut-off value of renal cortical stiffness of <27.3 kPa for detection of eGFR < 60 mL/min/1.73 m^2^ with 80% sensitivity and 85% specificity (AUC = 0.811, *p* < 0.0001) (Figure 7), a cut-off value of <26.9 kPa for detection of eGFR < 45 mL/min/1.73 m^2^ with 82.6% sensitivity and 74% specificity (AUC = 0.789, *p* < 0.0001) (Figure 8), and a cut-off value of <23 kPa for detection of eGFR < 30 mL/min/1.73 m^2^ with 88.8% sensitivity and 75.6% specificity (AUC = 0.852, *p* < 0.0001) (Figure 9).

Univariate and multivariate statistical analyses were used to examine the relationships between the median 2D SWE PLUS measures and the following parameters: age, gender, height, weight, BMI, transplanted kidney length, time from transplant, deceased or living donor, immunosuppressive treatment, complete blood count, urea, creatinine, sodium, potassium, calcium, phosphorus, PTH, uric acid, urine culture, VSH, C reactive protein, cholesterol, triglycerides, proteinuria/24 h AST, ALT, total bilirubin, and history of hypertension, diabetes, or previous glomerular disorders.

Univariate analysis showed that urea (*p* = 0.007), eGFR (*p* < 0.001), median Vi PLUS (*p* < 0.001), and median depth (*p* < 0.001) were independently associated with 2D SWE PLUS measures.

In multivariate regression analysis, only eGFR (*p* = 0.0001), median Vi PLUS (*p* = 0.0015), and median depth (*p* = 0.0018) were independently associated with 2D SWE PLUS measures (*p* < 0.0001).

Univariate and multivariate statistical analyses were used to examine the relationships between the Vi PLUS values and the following parameters: age, gender, height, weight, BMI, transplanted kidney length, time from transplant, deceased or living donor, immunosuppressive treatment, complete blood count, urea, creatinine, eGFR(*p* = 0.0180), sodium, potassium, calcium, phosphorus, PTH, uric acid, urine culture, VSH, C reactive protein, cholesterol, triglycerides, proteinuria/24 h AST, ALT, total bilirubin, and history of hypertension, diabetes, or previous glomerular disorders. The only correlation found in the univariate regression was between eGFR and Vi PLUS. No statistically significant multivariate regression model was found.

## 4. Discussion

We found a positive correlation coefficient between eGFR and median measure of renal cortical stiffness expressed in kPa. When comparing with other transplanted kidney research, we found that four studies assessing renal allograft injury were performed using TE elastography, and they all showed increased stiffness as the CKD stage progressed [6,7,8,9]. VTQ performed in two studies showed opposing results. The first one published by Stock et al. 2011 [10] showed increased stiffness, and the other one performed by Syversveen et al. 2011 [11] revealed decreased stiffness in relationship with eGFR. We also found one study using 2D SWE technology performed by Grenier et al. 2012 [12] which revealed no correlation between renal graft cortical stiffness and eGFR.

A recent study published by Richard Barr in 2020 showed that the displacement curves clearly illustrate that elastography systems were unable to produce accurate stiffness estimates, most likely due to reverberation errors from the renal cortex or kidney anisotropy. However, with the release of updated software for the new Supersonic Image on Mach 30, this problem may be resolved, but there have been no studies published employing the updated renal algorithm [19].

The 2D SWE method was originally designed to detect liver fibrosis [20], and the current study demonstrated that this approach can also be utilized to assess renal transplant parenchymal stiffness, which according to our findings, accurately reflects CKD stages.

In contrast with other Supersonic Image studies [21,22], our results highlight a positive correlation between decreased cortical stiffness as assessed with YM and the presence of CKD. The majority of prior investigations on native kidneys used acoustic radiation force impulse (ARFI) technology and reported considerably lower stiffness values in the CKD population using biopsy, GFR, serum creatinine, and scintigraphy as markers for impaired kidney function [23,24,25]. However, some studies identified no link between them and biopsy-proven fibrosis, CKD, or eGFR [26,27].

The meta-analysis of Hwang et al. 2021 [28] determined the technical performance of ARFI for evaluating renal parenchymal stiffness. The percentages of technical faults and intrasubject correlation coefficient indicated good agreement in both native and transplanted kidneys, but also that the region of interest location represented a significant factor of heterogeneity in transplanted kidneys. Our study shows good intra-operator agreement for both 2D SWE PLUS and Vi PLUS measures.

We obtained a cut-off value of renal cortical stiffness of <27.3 kPa for detection of eGFR < 60 mL/min/1.73 m^2^, which is in contrast with other studies. Radulescu et al. (2018) [21] obtained a cut-off value of 22.95 kPa for predicting CKD, and Samir et al. (2015) [22] obtained a cut-off value of 5.3 kPa when differentiating healthy kidneys from CKD. These discrepancies may be explained by the different groups: native kidneys (healthy and with CKD) versus kidney transplants, the anisotropy of the kidney, and the lack of a standardized modality for measuring stiffness.

Our study found no link between estimated stiffness measures and the presence of diabetes. Previous ones found significant differences in shear wave values in ARFI examinations between diabetes with microalbuminuria and diabetes with macroalbuminuria [29], or between different stages of diabetic nephropathy [23], leading to the conclusion that elastography could be a promising tool for detecting early kidney involvement in diabetes.

When analyzing the factors potentially influencing cortical stiffness estimation, no correlation between BMI and the median measures of 2D SWE PLUS was found. The influence of BMI on kPa estimates was also investigated in previous studies [21,22,25,30,31], and the majority found a correlation. In contrast, the median depth at which the evaluations were performed was independently associated with 2D SWE PLUS measures. This aspect was also pointed out in previous research [22,25,26]. Perhaps the variations of factors influencing the readings are due to the kidney grafts’ proximity to the skin. A recent meta-analysis of renal elastography investigations demonstrated that patients with a lower eGFR had lower kidney stiffness on average. However, even though renal elastography appears to be a promising method for monitoring CKD progression, studies up to date reveal great heterogeneity [32]. As intrarenal blood flow decreases as fibrosis progresses, alterations in renal perfusion may influence renal stiffness and explain some inconsistencies in results. As a consequence, a decrease in renal blood flow may be the cause of stiffness reduction as CKD progresses, and it may have a greater impact on stiffness than renal fibrosis [33].

Inflammation is a key factor in the progression of fibrosis [34]. Vi PLUS allows users to seek information regarding tissue shear wave dispersion, which can be used to measure viscosity indirectly [35].

Our study presents a positive correlation between eGFR and median Vi PLUS measures and between median measures of Vi PLUS and 2D SWE PLUS. Only a few studies have used non-invasive methods to assess tissue viscosity. Deffieux et al. (2015) were the first to use this imaging technique to evaluate liver viscosity [36]. The findings revealed that viscosity had a lower predictive value for fibrosis staging than hepatic stiffness and that it was only a marginal predictor of disease activity and steatosis levels. Sugimoto et al. (2018) [35] observed that the fibrosis stage was strongly associated with shear wave speed and that lobular inflammation grade was significantly connected to dispersion slope. Elasticity outperformed viscosity in predicting the stage of fibrosis, but viscosity outperformed elasticity in predicting the degree of necroinflammation [35]. We found no link between viscosity and inflammation in patients (correlation coefficient between Vi PLUS and C reactive protein or ESR were not statistically significant). We can speculate that viscosity measures that show acute inflammation may be beneficial in the assessment of acute rejection, but none of the individuals in our sample who underwent evaluation reported acute rejections. Given the fact that this was the first research to focuses on it, further investigation is needed to establish its role in clinical practice.

Non-invasive approaches such as 2D SWE PLUS and Vi PLUS will never be able to match the diagnostic strength of the gold standard, which is represented by renal allograft biopsy, and are unlikely to replace histology for determining fibrosis and loss of renal function after transplantation; but the most promising and appealing use would be the ability to track changes in allografted parenchymal structure over time. A progressive decrease in parenchymal stiffness over consecutive 2D SWE PLUS and Vi PLUS measures may help to define those individuals who may benefit most from a renal biopsy, even while serum creatinine levels are steady. Whether to perform a protocol biopsy can often be debated.

Renal biopsy has been shown to be a useful method for diagnosing subclinical allograft failure, particularly in cases of chronic allograft fibrosis or calcineurin inhibitor toxicity [37]. However, because of the risk of numerous complications, patients refuse the procedure. 2D SWE PLUS and Vi PLUS, on the other hand, are quick, noninvasive methods for assessing fibrosis, and have high patient acceptance, good reproducibility, and instantaneous results. The current work, along with other published studies [12], has demonstrated that it could properly identify stiffness modifications in relation with eGFR in kidney allografts. Patients with increasing interstitial fibrosis who would benefit most from a biopsy could be established using a longitudinal assessment of parenchymal stiffness [6,7,9].

There are a few limitations to our study that should be mentioned: the study’s small number of patients (the limited number of patients with G1T and G5T KDIGO stages), the lack of correlations between analyzed clinical data (may have resulted from the former), the magnitude of fibrosis proven by kidney biopsy, not taking into consideration renal perfusion problems, and the analysis of a limited number of factors that could influence cortical stiffness.

## 5. Conclusions

Assessment of renal allograft stiffness and viscosity may prove to be an effective method for identifying patients with chronic allograft injury and could prove to be a low-cost approach to provide additional diagnostic information on kidney transplanted patients.

## Figures and Tables

**Figure 1 jcm-11-04370-f001:**
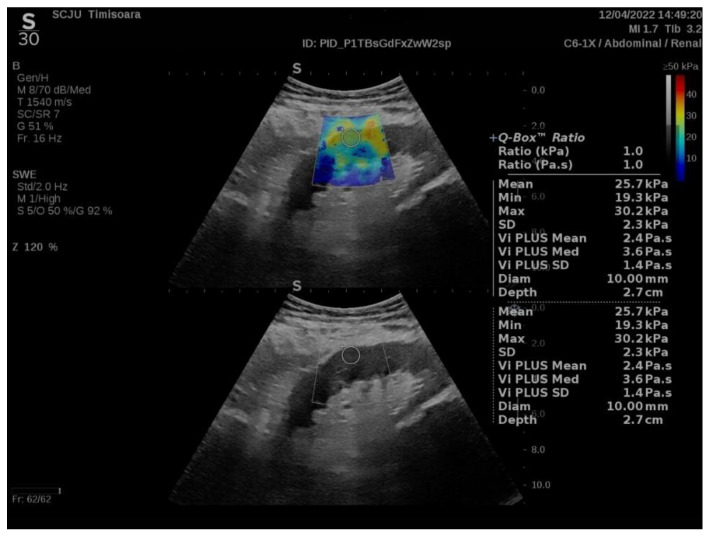
Quantitative elasticity map of the middle portion subcapsular cortex of a transplanted kidney.

**Figure 2 jcm-11-04370-f002:**
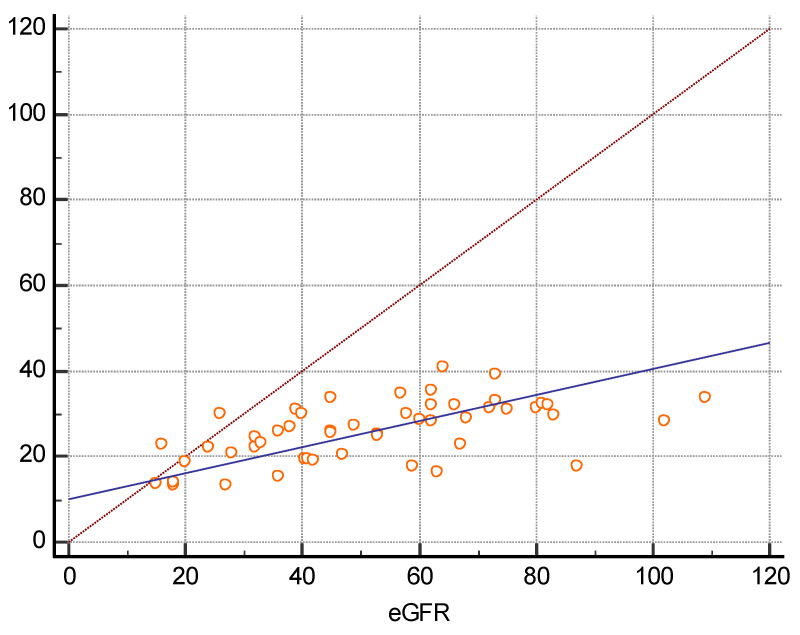
Scatter diagram between Median 2D SWE PLUS measures and eGFR.

**Figure 3 jcm-11-04370-f003:**
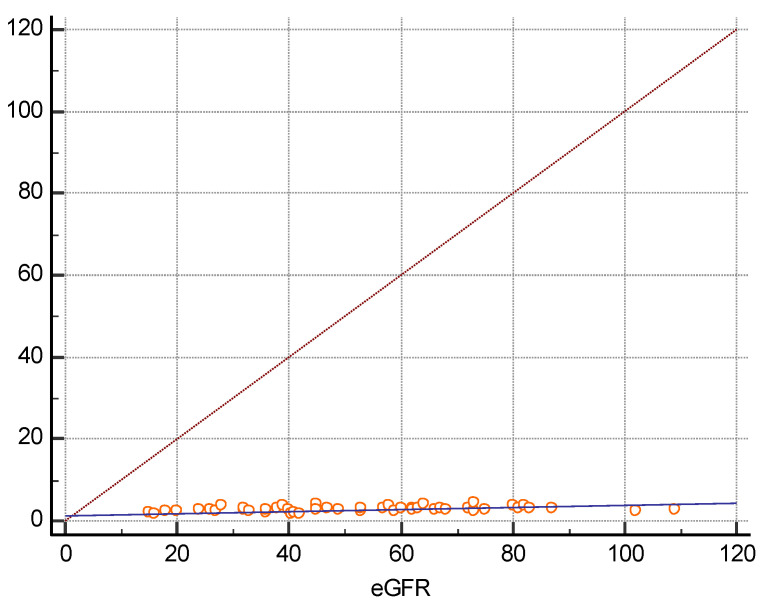
Scatter diagram between Vi PLUS measures and eGFR.

**Figure 4 jcm-11-04370-f004:**
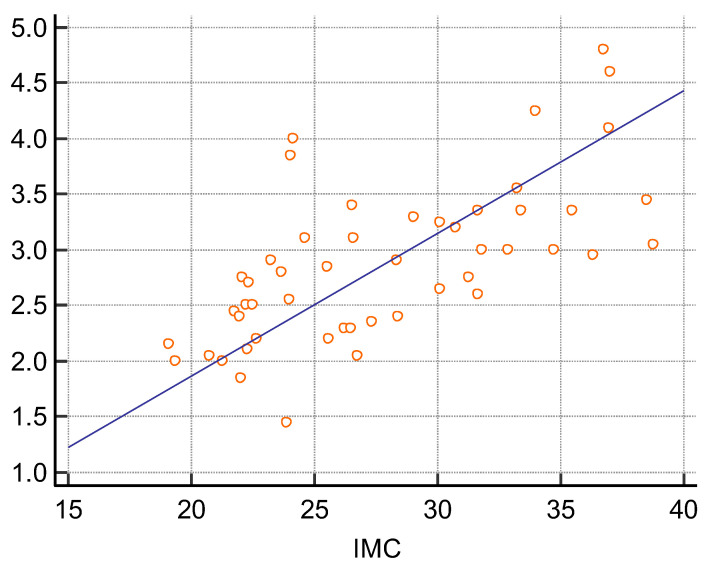
Scatter diagram between median depth of measures and IMC.

**Figure 5 jcm-11-04370-f005:**
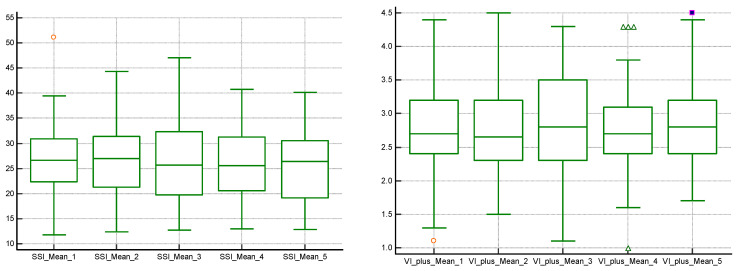
Box-and-whisker plots for ICC for 2D SWE PLUS and Vi PLUS measures.

**Figure 6 jcm-11-04370-f006:**
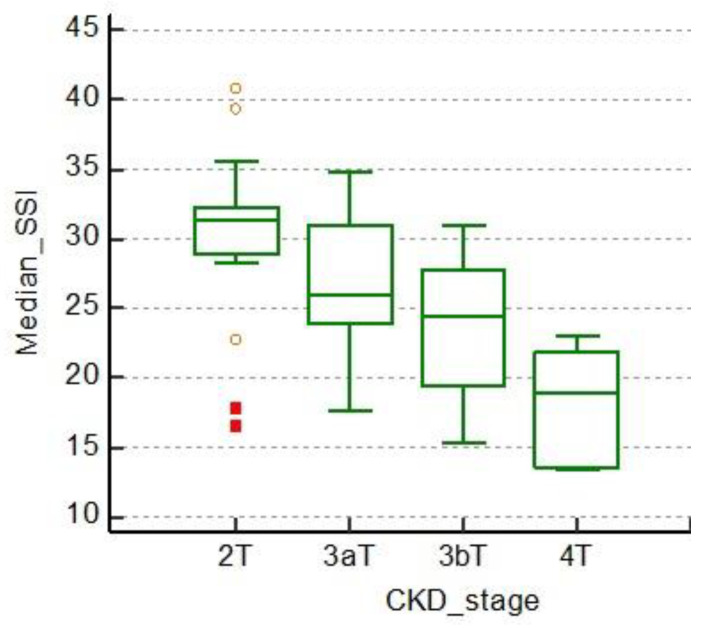
Multiple comparison graphs between CKD stages and median values of 2D SWE PLUS measures.

**Figure 7 jcm-11-04370-f007:**
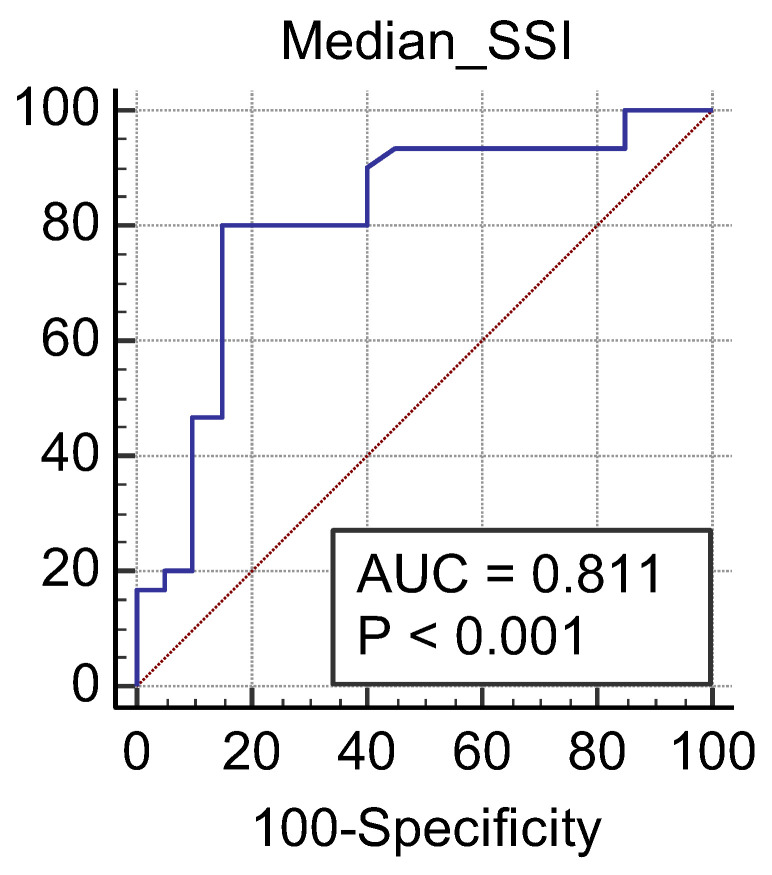
Performance of 2D SWE PLUS for predicting eGFR < 60 mL/min/1.73 m^2^.

**Figure 8 jcm-11-04370-f008:**
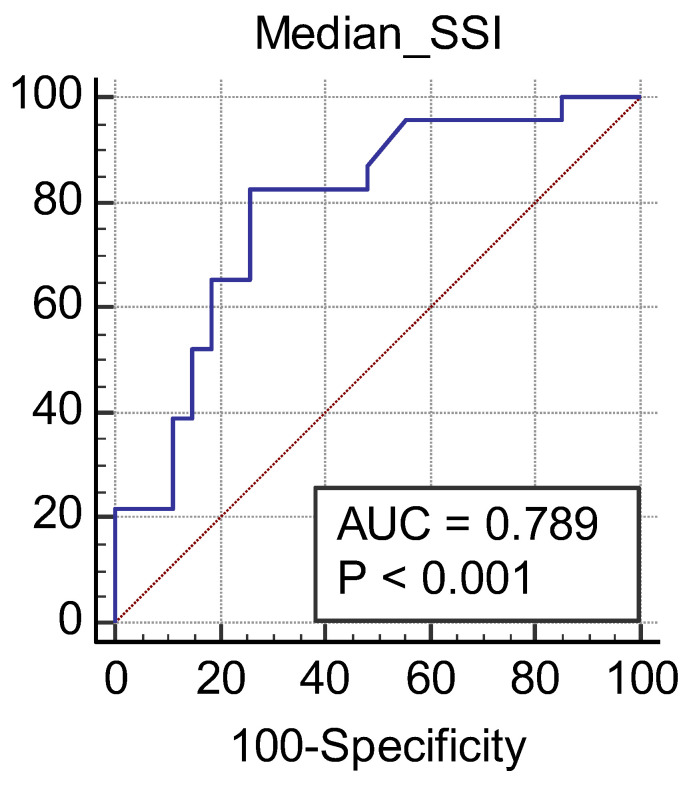
Performance of 2D SWE PLUS for predicting eGFR < 45 mL/min/1.73 m^2^.

**Figure 9 jcm-11-04370-f009:**
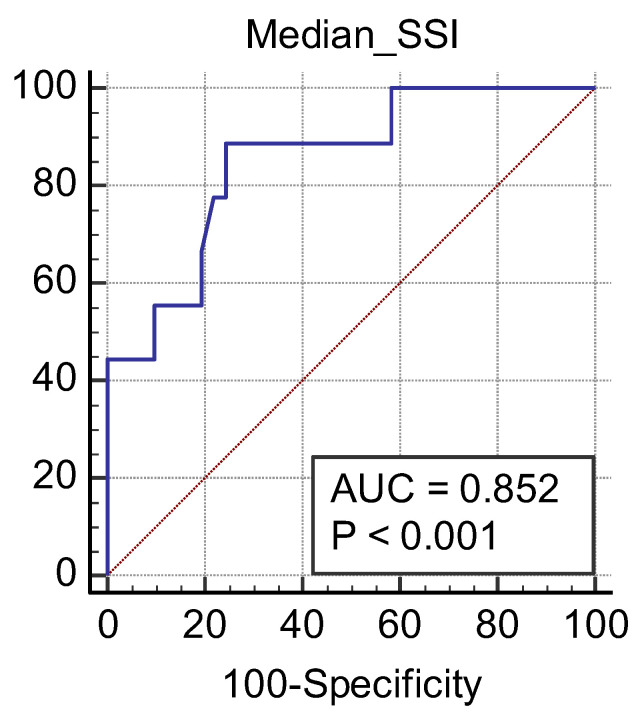
Performance of 2D SWE PLUS for predicting eGFR < 30 mL/min/1.73 m^2^.

**Table 1 jcm-11-04370-t001:** Mean values of 2D SWE PLUS measures in different CKD stages.

CKD Stage	Number of Subjects	Mean2D SWE PLUS Measures ± Standard Deviation
**2T**	18	30.18 ± 6.21 kPa
**3aT**	9	26.73 ± 5.64 kPa
**3bT**	13	24.03 ± 4.83 kPa
**4T**	7	17.95 ± 4.3 kPa

**Table 2 jcm-11-04370-t002:** Correlations between patients’ characteristics and median 2D SWE PLUS measures (n = 50).

Gender	34 Men (68%) 16 Women (32%)	Correlation Coefficient with 2D SWE PLUS r	Significance Level *p*
**Height (meters)**	1.70 ± 0.08	0.2679	*p* = 0.0599
**Weight (kilograms)**	81.3 ± 18.2	−0.01267	*p* = 0.9304
**BMI**	27.8 ± 5.5	−0.1738	*p* = 0.2274
**Time from transplant (years)**	10.5 ± 5	0.09945	*p* = 0.4920
**Attained from a living related/deceased donor**	20 living related 30 deceased donor	-	-
**Kidney length (millimeters)**	116.04 ± 14.3	0.1659	*p* = 0.2496
**eGFR (mL/min/1.73m^2^)**	52 ± 22.6	0.5699	*p* < 0.0001
**Hemoglobin (g/dL)**	13.3 ± 1.5	0.2487	*p* = 0.0816
**Hematocrit (%)**	41.2 ± 5	0.1078	*p* = 0.4563
**Urea (mg/dL)**	50 ± 20.9	−0.3766	*p* = 0.0070
**Uric Acid (mg/dL)**	6.94 ± 1.5	0.1078	*p* = 0.4563
**Cholesterol (mg/dL)**	202.8 ± 61	−0.1028	*p* = 0.5821
**Triglycerides (mg/dL)**	177.8 ± 80.4	−0.05210	*p* = 0.7845
**ALT (U/L)**	23.3 ± 15.4	0.3041	*p* = 0.0963
**AST (U/L)**	23.9 ± 12.8	0.1794	*p* = 0.3343
**Total Bilirubin (mg/dL)**	0.66 ± 0.2	−0.04079	*p* = 0.8534
**Sodium (mg/dL)**	135.97 ± 20.9	0.01393	*p* = 0.9348
**Potassium (mg/dL)**	4.37 ± 0.5	−0.2071	*p* = 0.2187
**C reactive protein**	14.9 ± 35.2	−0.2390	*p* = 0.1375
**History of- hypertension** **-diabetes** **-previous glomerular disease**	37 8 19		

Data are presented as numbers and percentages or means ± standard deviations.

## Data Availability

Not applicable.

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
