# Peer review of "Assessment of Renal Allograft Stiffness and Viscosity Using 2D SWE PLUS and Vi PLUS Measures—A Pilot Study"

_jcm, 2022, doi:10.3390/jcm11154370_

Round 1

Reviewer 1 Report

This manuscript well describe a cross-sectional study on renal transplant recipients where two new ultrasound-based technique (ShearWave Elastography and Viscosity Plane-wave UltraSound) are explored in order to define their suitability to estimate the stiffness of the graft and its correlation to eGFR.

The results seem promising although the small sample. 

I suggest rechecking the acronyms and to improve the paragraphs of the abstract.

Author Response

We would like to thank the reviewer for the kind words, valuable comments, and suggestions. We hope that all the changes that have been performed lead to an improvement of the manuscript.

All the modifications in the text have been performed using “Track Changes” and can be easily followed up.

We hope we could answer appropriately to all raised concerns.

Reviewer 2 Report

Promising data shown on a small study group, with necessity to expand beyond a pilot study.

Introduction

Every renal allograft undergoes chronic injury and a noninvasive method to track the progression of fibrosis is a tempting diagnostic option. However, the major clinical challenge is to diagnose an early acute rejection. The current data on elasticity/viscosity assessement is such conditions would enrich the content of the introduction.

Methods

The number of patients is low, but the Authors admit that in the study limitations. However, statistical analysis taking into account CKD stage 1 (2 patients) and stage 5 (1 patient) as separate subgroups is unjustified (Fig.6).

Lack of correlations between the majority of analyzed clinical data may result from the small number of patients (Tab.2) and such comment should be put into the limitations of the study.

Results

Page 4 lines 168-169

Is the term "mean median" an appropriate one?

Discussion

The Authors found elasticity more useful in the assessment of fibrosis, whereas viscosity turned out less effective in tracking the inflammatory changes. This is not surprising taking into account the characteristics of chronic allograft injury. Would the Authors dare hypothesising on the potential role of viscosity measurements in acute rejection conditions, when inflammation takes over?

Author Response

Responses to Reviewer 2

 We would like to thank the reviewer for the kind, valuable comments, and suggestions. We tried to answer every raised comment and we hope that all the changes that have been performed lead to an improvement of the manuscript.

As follows we summarize all the comments of the reviewer with the answers provided by us. All the modifications in the text have been performed using “Track Changes” and can be easily followed up.

Reviewer comment:

Every renal allograft undergoes chronic injury and a noninvasive method to track the progression of fibrosis is a tempting diagnostic option. However, the major clinical challenge is to diagnose an early acute rejection. The current data on elasticity/viscosity assessment is such conditions would enrich the content of the introduction.

Response

The reviewer is perfectly right,  the major clinical challenge would be to diagnose an early acute rejection, we added the suggestion in the introduction section with” track changes”.

Reviewer comment:

The number of patients is low, but the Authors admit that in the study limitations. However, statistical analysis taking into account CKD stage 1 (2 patients) and stage 5 (1 patient) as separate subgroups is unjustified (Fig.6).

Lack of correlations between the majority of analyzed clinical data may result from the small number of patients (Tab.2) and such comment should be put into the limitations of the study.

Response

We would like to thank the reviewer for the excellent comments, we excluded from Fig. 6 and Table 1: CKD stage 1 and CKD stage 5 patients.

We also included in the limitatons section of the study the lack of correlation between the majority of analyzed clinical data and that it may result from the small number of patients.

Reviewer comment:

Is the term "mean median" an appropriate one?

Response

We changed it to ” the arithmetic mean of the median of the five measures"

Reviewer comment:

The Authors found elasticity more useful in the assessment of fibrosis, whereas viscosity turned out less effective in tracking the inflammatory changes. This is not surprising taking into account the characteristics of chronic allograft injury. Would the Authors dare hypothesising on the potential role of viscosity measurements in acute rejection conditions, when inflammation takes over?

Response

We would like to thank the reviewer as well for the last paragraph. We would certainly dare hypothesising on the potential role of viscosity measurements in acute rejection conditions when inflammation takes over, we included this aspect in the discussion section as well.